# A Scientific Approach to Conscious Experience, Introspection, and Unconscious Processing: Vision and Blindsight

**DOI:** 10.3390/brainsci12101305

**Published:** 2022-09-27

**Authors:** Reinhard Werth

**Affiliations:** Social Pediatrics and Adolescent Medicine, Ludwig-Maximilians-University of Munich, Haydnstr. 5, D-80336 München, Germany; r.werth@lrz.uni-muenchen.de; Tel.: +49-(0)-1733550232

**Keywords:** blindsight, conscious experience, vision, behaviorism, visual system, superior colliculi, pulvinar

## Abstract

Although subjective conscious experience and introspection have long been considered unscientific and banned from psychology, they are indispensable in scientific practice. These terms are used in scientific contexts today; however, their meaning remains vague, and earlier objections to the distinction between conscious experience and unconscious processing, remain valid. This also applies to the distinction between conscious visual perception and unconscious visual processing. Damage to the geniculo-striate pathway or the visual cortex results in a perimetrically blind visual hemifield contralateral to the damaged hemisphere. In some cases, cerebral blindness is not absolute. Patients may still be able to guess the presence, location, shape or direction of movement of a stimulus even though they report no conscious visual experience. This “unconscious” ability was termed “blindsight”. The present paper demonstrates how the term conscious visual experience can be introduced in a logically precise and methodologically correct way and becomes amenable to scientific examination. The distinction between conscious experience and unconscious processing is demonstrated in the cases of conscious vision and blindsight. The literature on “blindsight” and its neurobiological basis is reviewed. It is shown that blindsight can be caused by residual functions of neural networks of the visual cortex that have survived cerebral damage, and may also be due to an extrastriate pathway via the midbrain to cortical areas such as areas V4 and MT/V5.

## 1. Introduction: The Theoretical Background

The psychologist J.B. Watson considered psychology as an experimental science. In Watson’s view, introspection does not belong to its methods, and does not contribute to scientific knowledge. Watson was convinced that “the time seems to have come when psychology must discard all reference to consciousness; when it need no longer delude itself into thinking that it is making mental states the object of observation” [1] (p. 263). Radical behaviorist positions have also been held by philosophers, among which the most recognized advocates are Ludwig Wittgenstein [2] and Gilbert Ryle [3]. Wittgenstein argues that the terms of an intersubjectively valid, objective language cannot refer to subjective sensations to which only those who have these sensations have access. In this case, there is no way for the community of speakers to decide whether the relation between a sensation and the term that designates it is correct. Since such an objective criterion for the correctness of this relation is missing, it is not possible to speak of “correct” and “incorrect” here. These terms belong to a private language, that is not intersubjectively understandable. Wittgenstein compared conscious subjective experiences with a beetle in a box, where each person owns a box and claims to know what a beetle was by looking into his own box. However, no one can look into another person’s box. Then, it was possible that everyone had something else in his box or his box could also be empty. The word “beetle” does not designate an object, and can be eliminated from the language game [2] (§293) [4]. Similarly, the English philosopher Gilbert Ryle [3] regards consciousness as a ghost in the human machine and claims that no person has privileged access to himself. Ryle claims that what one can find out about oneself is based on the same methods as what one finds out about other people. However, the claim that no one has privileged access to himself contradicts everyday observations. Imagine that a person P is asked to press one of three keys, where gains and losses are identical for each key-press. From an observer’s perspective, each key-press has the same a priori probability. If a person other than P attempts to predict which key P will press next, s/he needs to observe P’s reaction in numerous similar situations. However, it is hardly possible to predict with certainty which key P will press next. Nevertheless, we can state in an objectively verifiable manner that P can predict with absolute certainty which key s/he will press next. P never needs to have performed such a task before, and s/he never needs to have observed his/her behavior in such a task. There is no doubt that P has privileged access to his/her decision which key s/he will press next.

Positivism and the resulting behaviorism advocated by philosophers of the Vienna Circle can be considered one of the cornerstones of the philosophy of science. From the perspective of Rudolf Carnap, one of the most prominent advocates of this philosophical school, every proposition of psychology can be formulated in physical language. Carnap subsumes the philosophical behaviorism of the Wiener Circle with the words “The introspective statements of a psychologist are not, in principle, to be interpreted differently than the statements of his experimental subject about whom he reports. Additionally, the statements of an experimental subject are not, in principle, to be interpreted differently than his other voluntary or involuntary movements, though his speech movements may under favorable circumstances be regarded as especially informative. Again, the movements of the speech organs and of the other experimental subject’s body parts are not, in principle, to be interpreted differently than the movements of an animal.... The movements of an animal are not, again in principle, to be interpreted any differently than those of a voltmeter.... Finally, the movements of a voltmeter are not, in principle, to be interpreted differently than the movements of a raindrop...” [5] (p. 140). A few years later, Carnap recognized that the “...psychological movement of Behaviorism had, on the one hand, a very healthful influence because of its emphasis on the observation of behavior as an intersubjective and reliable basis for psychological investigations, while, on the other hand, it imposed too narrow restrictions. First, its total rejection of introspection was unwarranted. Although many of the alleged results of introspection were indeed questionable, a person’s awareness of his own state of imagining, feeling, etc., must be recognized as a kind of observation, in principle not different from external observation, and therefore as a legitimate source of knowledge, though limited by its subjective character” [6] (pp. 70–71). Additionally, B.F. Skinner, one of the outstanding American psychologists of the 20th century, father of operant conditioning and founder of a school of thought he designated “radical behaviorism”, did not deny the existence of subjective sensations, and he did not consider introspective reports merely as verbal behavior. Skinner advocated only that subjective states do not contribute to the analysis of behavior, and that they are not suitable for explaining behavior [7,8].

Terms that designate conscious, subjective experiences do not designate behavior and environmental conditions under which behavior occurs, and they cannot be defined by describing behavior and environmental conditions. This raises the question of whether conscious sensations can be introduced into the language of science at all. One objective of the present paper is to demonstrate how terms that designate conscious experiences can be introduced into science and how conscious experiences and unconscious processing be can be distinguished from each other scientifically. To achieve this, the concepts “conscious experience” and “unconscious processing” must be translated into a language of mathematical logic that clarifies its semantic nature and does not lead to contradictions or obscure assertions. The distinction between “conscious”, “unconscious” and “reduced conscious experience” is demonstrated with the example of the processing of visual stimuli in the presence of different levels of conscious visual experience. The literature on vision with reduced visual experience (blindsight) will be reviewed. The neurobiological foundations of blindsight are reviewed on the basis of the literature and our own studies of adult patients with injury to the occipital lobe, children with one or both occipital lobes missing, children after hemispherectomy, and children without the telencephalon. The review is based on several thousand publications about the anatomy, physiology, and neuropsychology of the visual system in humans which were available in the Max-Planck-Institute for Psychiatry, the Bavarian State Library, the Library of the Medical Faculty of the University of Munich, Pubmed, Science Direct, Psycnet or other internet-based databases, which the author collected over about 40 years up to the year 2022. A total of 190 studies that were considered the most relevant to the questions posed in the present review were included. 

## 2. Psychological Terms Understood as Theoretical Concepts

Carnap realized that many major psychological concepts cannot be defined by terms that designate observational entities such as observable behavior under given observable circumstances [9]. For example, if we attempted to define the term “arachnophobia” by the statement “P has arachnophobia” it would mean “whenever P sees a spider, P displays fear reactions R”, this led, for formal logical reasons, to the absurd consequence that one could diagnose arachnophobia if this person has never seen a spider. Carnap therefore suggested that terms introduced by specifying a certain reaction under certain conditions should be introduced by so called “reduction sentences” [9]. A reduction sentence by which the term “arachnophobia” is introduced has the following logical form: “If a Person P sees a spider, then, if P has arachnophobia, P displays fear reaction R, and if P has no arachnophobia, P does not display fear reaction R” [9] (p. 440), [10]. In terms of formal logic: S(P) → (A(P) ↔ R(P)), whereby S(P) represents “Person P sees a spider”, A(P) “P has arachnophobia”, R(P) “P displays fear reaction R”, → is the logical sign for implication, and ↔ is the logical sign for equivalence. However, people with arachnophobia do not display only a single reaction under one environmental condition. The Diagnostic and Statistical Manual of Mental Disorders (DSM5), [11] (pp. 197–198). specifies different reactions that occur under different conditions for the diagnosis of a specific phobia. Therefore, we can introduce the concept of arachnophobia not with a single reduction sentence, but with several reduction sentences. Over time, new reduction sentences can be added and others may be abandoned, and new psychobiological statements can further specify a particular type of phobia. This introduction of a term violates all conditions that apply to a definition and should therefore not be regarded as an “operational definition” as is often the case in psychological texts. Carnap [6,12] designated these terms as “theoretical concepts”. Theoretical concepts are linked to terms that designate observables, but are not defined by them. What has been said so far is also true for all terms introduced by specifying the conditions under which a given reactions must occur in order for us to say that a term can be applied. Terms, such as “seeing”, “hearing”, “pain”, “fear”, “restlessness”, “depression”, “euphoria”, etc. are introduced by behavior under given environmental conditions. If a person sees something, hears something, feels pain, fear, or restlessness, or is depressed or euphoric, these can only be known by investigating his/her behavior under given environmental conditions. These terms can be regarded as theoretical terms that, like theoretical terms in physics, need not refer to introspectively accessible conscious experiences. However, we assume that many of these terms designate conscious experiences. Experimental findings demonstrate that a distinction between conscious experiences and unconscious processing is inevitable. Studies on conscious visual experience and unconscious visual processing of stimuli in an apparently blind visual field, termed “blindsight”, is an example that demonstrates the importance of a distinction between conscious experience and unconscious processes. The questions are how this distinction can be achieved and how these conscious experiences can be accounted for in a scientifically accurate manner. 

## 3. Conscious Visual Experience and Unconscious “Vision” (Blindsight)

If we assess a person’s luminance difference threshold, the reduction sentences indicate under which visual stimulation which response should occur for a stimulus to be considered as “seen”. Then, the term “a subject has seen the stimulus” is introduced by appropriate reduction sentences. In scientific practice, it is already known how to measure an incremental and a decremental threshold. If we assess the visual field using visual perimetry, we can instruct the subject to press a button whenever s/he sees a light spot; by “seeing” we mean the “conscious experience of seeing”. Thus, we presuppose an everyday understanding of the term “conscious seeing” in the subject´s instruction. However, seeing does not always presuppose the presence of conscious visual experience. Visual stimuli can also be registered and localized by the human visual system in the absence of conscious experience. Pöppel, Held and Frost [13] were the first to demonstrate that patients who were apparently blind in an area of their visual field after damage to the occipital lobe, and who also asserted that they could not see anything at all in the affected visual field, could nevertheless locate stimuli correctly when asked to guess the location of stimuli, and to point to the location where they guessed the presence of the stimuli. Weiskrantz et al. [14] termed this phenomenon “blindsight”. Subsequent experiments (e.g., [15,16,17,18,19,20,21,22,23,24,25,26,27,28,29,30]) not only confirmed that patients are able to locate stimuli in an apparently blind area but also that they can distinguish shapes, colors, and objects in an area that is perimetrically blind in which patients assert that they have no conscious visual experience. Zihl and von Cramon [19] demonstrated that patients who had performed several hundred practice trials were able to register the presence or absence of a light stimulus that was presented in the apparently cerebrally blind visual field even though they denied any conscious visual experience. A patient examined by Zihl and Werth [21,22] had a right homonymous hemianopia due to a stroke of the left cerebral hemisphere. The right half of the visual field of both eyes was blinded. The visual field of the patient was assessed using the Tuebingen perimeter. The patient looked into the perimetric hemissphere (diameter 66 cm) with his head stabilized and directed his gaze to a point in the center. Light spots were presented alternately at five different locations on the horizontal meridian in the perimetric hemisphere for 100 ms so that all light spots were projected into the blind visual hemifield of the retina. In all experiments scattering light was measured, and its influence was excluded [31]. An acoustic signal indicated when a light spot was present. Since the patient claimed not to see anything in the right visual hemifield, he was asked to guess where the light spot was located and to look at the location where he guessed the presence of the light spot. The presentation time of the light spots was so short that they had disappeared by the time the eyes began moving to the guessed location. Surprisingly, there was a clear correlation between the location to which the patient looked and the location where the light spots appeared. If this ability to guess the locations was not spontaneous it could be trained in a short period of time [20]. Weiskranz et al. [32] demonstrated that a patient with damage to the primary visual cortex (V1) could discriminate the direction of stimuli when the possible influence of scattering light was also excluded. When the contrast was increased and the stimuli moved with high velocity, the patients became aware of the stimuli. Ffytche and Zeki [33] showed that in direction of motion experiments that visual awareness was more likely when stimuli were moving compared to static stimuli. Whether a patient became aware of a moving stimulus depended only on the magnitude of activation of the cortex in area V5 [34]. The phenomenon that stationary stimuli were not detected in a visual field affected by a lesion of the occipital cortex, whereas moving stimuli were consciously seen, was first described by Riddoch [35]. This kind of awareness of some stimuli in a visual field where the patient is unaware of other stimuli presented in the affected visual field, was termed as “type 2 blindsight”. The case of type 2 blindsight, in which the stimulus is not seen but a non-visual sensation is reported, (for example, the feeling that something is present when the stimulus is presented in the blind field) must be distinguished from the Riddoch phenomenon, which has also been designated as “type 2 blindsight” [36,37,38,39,40,41,42,43,44,45,46,47,48]. 

## 4. How to Distinguish between Conscious Vision, Non-Visual Experience and Blindsight?

In many studies on visual abilities in an apparently blind visual field, the patients were asked, only after the experiment, whether they had a visual experience. Even if the patients claim not to have seen anything, one cannot be sure that they did not have a subjective experience during the experiment. Patients may also not direct their attention to any weak sensation whatsoever, and consequently, may not remember any such sensation at the end of the experiment. Therefore, it is more appropriate to ask the patients immediately after each experimental trial if they only guessed, whether they could see something or whether any sensation occurred. However, unusual visual experiences can occur after brain damage, for which everyday language has no terms. Patients are often unsure if such experiences can be designated as “seeing” and, therefore, deny the question of whether they saw something. Instead of merely relying on the patient’s statement that they had no visual experience and were merely guessing, patients can be asked to indicate their level of confidence on a scale [49]. The response with which a person indicates the presence or absence of a stimulus, its location, shape, color, etc., will be named “first-order response”, the response with which s/he indicates what confidence s/he has in the correctness of the first-order response, whether s/he only guessed or any conscious sensation occurred, will be named “second-order response”. For example, Mazzi et al. [50,51] used a four-level scale to examine a patient’s subjective experience. A patient who had developed right homonymous hemianopia due to a hemorrhagic stroke was able to discriminate the orientation, color and contrast of a stimulus in her blind visual hemifield when she declared that she responded only by guessing. The patient was asked to rate her subjective experience as either “no visual experience”, “brief glimpse”, “almost clear visual experience”, or “clear visual experience”. The patient was unable to discriminate between two stimulus orientations, motion direction or between two colors when she reported no visual experience or perceiving a small glimpse. Contrast discrimination was only possible at instances when the patient experienced a small glimpse or when she saw the stimuli almost clearly. This experiment demonstrates that it is not sufficient to ask patients whether their responses were made on the grounds of guessing or whether the stimuli were visible. A rating procedure yields more information about the patients´ subjective experiences, but still does not exclude the case where the stimuli elicit a conscious experience which the patients do not term “seeing”, “perceiving a glimpse”, or having any other conscious experience for which the ordinary language has no words. However, it has already been demonstrated earlier [36] that there are more accurate ways to exclude feelings elicited in blindsight experiments. This study has shown that the degree to which a patient can introspectively recognize his ability of blindsight depends on the experimental conditions. The subjective experience of a 34-year-old hemianopic patient (HU) who was involved in a car accident was extensively investigated in a blindsight experiment [36]. A CT-scan revealed damage to the geniculostriate projection and its target areas in the left cerebral hemisphere. The lesion caused a right homonymous visual field defect. During the examination, the patient looked into the bowl of a Tuebingen perimeter. Light stimuli (diameter: 69 min/arc; luminance 126.7 cd/m^2^; background luminance: 3.2 cd/m^2^) were presented for 500 ms at a location within the blind area of the patient’s visual field. Fixation was controlled through the telescope of the perimeter. An acoustic signal indicated the beginning of a time interval in which either a light stimulus was provided or no light stimulus was present. Intervals in which a light stimulus was present and intervals in which no light stimulus appeared alternated in random order. In 50% of the acoustically marked intervals, a light stimulus was present. No light stimulus was present in the remaining acoustically marked intervals. The patient assured that he could not see anything in the blind area; hence he was asked to guess whether or not a light spot was present. Although the patient affirmed that he had never seen a light stimulus, he guessed correctly in 99% of the 290 experimental trials. The scattering light was measured to exclude its effect. In addition, the stimuli were presented in an area of the visual field that was completely blind and where no unconscious visual processing took place. In a subsequent experiment (Experiment 2), which will also be called “second order experiment”, the same stimuli were presented under the same conditions as in Experiment 1 which will also be called “first order experiment”. Experiment 2 differed from experiment 1 in that the stimuli were presented alternately in a pseudorandom order in 50% of the trials at a location where no processing of stimuli was discovered, and in 50% of the trials at a location where blindsight was demonstrated. The patient was asked to rate his ability to detect the stimulus on a three-level scale. If he was absolutely convinced that he could not detect a stimulus, he was asked to say “no”. If the patient even had the faintest idea that he could detect the stimulus, he should say “weak”, and if he was somewhat sure that he could detect the presence of a stimulus he should say “good”. The latter answer never occurred.

In 61% of 300 trials, the patient’s second-order responses (i.e., the verbal responses “no” and “weak”) were correct. This means that he answered “no” if the stimulus was presented at a location of the visual field where no processing of stimuli had been discovered in a previous experiment, and he answered “weak” if the stimuli appeared at a location where blindsight had been demonstrated in a previous experiment. When the patient was asked after the experiment which subjective experiences he had during the experiment, he stated that he had not even a faint feeling of the presence or absence of a stimulus. The control experiment was identical to the second-order experiment except that a stimulus was always presented outside the patient´s visual field. The difference between the result of the second-order experiment and the result of the control experiment was significant (chi-square test: *p* ≤ 0.005). Although a chi-square test showed the result to be significant, this should be interpreted with caution. The percentage (61%) can only be regarded as the degree to which the patient was aware that he was able to distinguish the presence or absence of a stimulus, if it is presupposed that this result did not come about by chance. Therefore, repeated control experiments must demonstrate that subjects cannot achieve a result of 61% correct responses when they generate a sequence of the answers “no” and “weak” in the presence or absence of a stimulus that does not influence their responses. This is, for instance, the case if the stimuli are always presented outside the visual field. Only if it is demonstrated that control subjects cannot achieve a result of 61% correct responses in repeated control experiments, the result of the above described second-order experiment can be regarded as being affected by the presence of the stimulus on a location where no blindsight was possible or on a location where blindsight was possible. Only then the result of 61% can be regarded as a very low level of awareness. 

In a second-order experiment, the level of awareness may have a value between 50% and 100%. Therefore, it seems more reasonable to assume that there is a continuous transition between visual processing without awareness (blindsight) and conscious visual perception instead of drawing an arbitrary boundary between type 1 and type 2 blindsight. To investigate the patient´s ability to rate his capacity to process visual stimuli in his apparently blind field in more detail, Experiment 2 was slightly modified and repeated. This experiment was identical to that previously described. The stimuli were again presented alternately in random order in 50% of the trials at a location where no processing of stimuli was possible, and in 50% of the trials the stimuli appeared at a location where blindsight had been demonstrated. The patient was informed that a light spot would be present in every trial. Before the experiment, the light spot was presented at a location where processing of stimuli was possible (blindsight), and the patient was informed that he would be able to detect the presence or absence of the light spot under these conditions. The light spot was also shown at the location where no blindsight occurred, and the patient was informed that under these conditions he could not detect the presence or absence of the stimuli. The patient was asked to compare the trials and indicate the trial in which he felt it was more likely that he could detect the stimuli. In this experiment, the patient´s answer was 100% correct in 300 trials. Such an experiment has the advantage that the patient receives feedback about the sensations associated with the ability to distinguish between the presence and absence of the stimuli. If the presence or absence of a light spot is accompanied by different non visual sensations he can identify them, and he can recognize their significance. In experiments where the patient does not receive such feedback, different non-visual sensations may also be associated with the presence or absence of the light stimulus. In this case, the patient may not recognize that these sensations indicate the presence or absence of a stimulus.

The control experiment was identical to this second-order experiment except that no light stimulus was shown in any trial. The difference between the result of the second-order experiment and the result of the control experiment was significant (chi-square test: *p* ≤ 0.0001). Here what has already been stated above is true. Control experiments must demonstrate that the experimental result is not due to chance, and that the outcome of the second-order experiment differs significantly from the outcome of the control experiments. The result of such a second-order experiment (i.e., the degree to which a person can correctly state whether the stimulus was presented at a location of the visual field where s/he was able to discriminate between the presence or absence of a light spot in a first-order experiment or whether the stimulus was presented at a location of the visual field where s/he was unable to discriminate between the presence and absence of a light spot in the first-order experiment) can be interpreted as the degree of introspection in his/her ability to detect the presence or absence of a stimulus. The first-order experiment investigates whether a patient can discriminate between the presence or absence of a stimulus. The second-order experiment investigates whether a person has introspective access to his/her ability to discriminate the presence or absence of a stimulus. The result of this experiment also can have values between 50% and 100%. This also suggests that we should not draw a sharp boundary between “no processing of stimuli”, “type 1 blindsight” and “type 2 blindsight”.

It is noteworthy to state that presenting a light spot at a location of the retina that does not lead to conscious perception or blindsight is not the same as presenting no light stimulus. A light stimulus at this location can elicit activation in a cortical area that leads to neither conscious experience nor blindsight [52]. If no stimulus is presented, no activation occurs. In the experiment described here, the difference between insufficient and sufficient activation was assessed.

The results of this study demonstrated that the diagnosis of “blindsight” cannot be made on the basis of the patients’ assertion that they don´t see anything and that they only have to guess. The necessity of a scientific introduction of the terms “conscious” and “unconscious” is demonstrated by the example of the insufficient attempt of Railo and Hurme [53] to characterize the terms “conscious” and “unconscious”. The authors write: “We use the term “conscious” vision to refer to visual perception that is accompanied by experiences that can be introspected by the subject. We use “unconscious” visual perception to refer to situations where the stimuli that the subject denies consciously seeing can nevertheless influence their behavior in some way” [53] (Section 2.1, first paragraph). An unclear term (conscious) cannot be introduced by characterizing it using another unclear terms such as “visual perception that is accompanied by experiences that can be introspected by the subject”. What is the scientific meaning of “visual perception that is accompanied by experiences” what is the meaning of “introspection”? To date, these terms are used in an unscientific, obscure way in scientific writing. Next it is shown how the concepts “conscious vision” and “blindsight” can be introduced in a logically and experimentally correct way. 

## 5. Introducing the Concepts “Introspection”, “Conscious Vision” and “Blindsight”

As already explained above, the term “conscious vision” must be understood as a theoretical term, introduced by reduction sentences that specify the responses in the second-order experiment under given experimental conditions. If a person is aware of his/her ability to discover the presence or absence of stimuli, shape, color, location, orientation, or direction of movement in the sense mentioned above, s/he has correctly evaluated his/her own abilities without being informed by the experimenter about the outcome of the experiment. In this case, we say that s/he has introspective access to his/her conscious experiences, and that s/he can consciously see the stimuli. These terminologies are common in the interpretation of neuropsychological results, but it misses any scientific justification, and behaviorist objections are still valid. This gives rise to the question, of how one can demystify “conscious experience” and introduce it logically into a contradiction-free language. Hence, the concept “conscious visual experience” must be translated into a language of mathematical logic that clarifies its semantic nature and does not lead to contradictions or obscure assertions.

If D(P(Ec, Sp, Sa)) represents “a Person P can, under experimental conditions Ec, discriminate between the stimuli Sp and Sa”, this is a theoretical term (disposition predicate) which denotes the disposition of a person to respond in a given way when certain experimental conditions are established. If I+(D(P(Ec, Sp, Sa))) represents “P can correctly evaluate his/her ability to discriminate under experimental conditions Ec between the stimuli Sp and Sa” this is also a theoretical term (disposition predicate) which denotes the disposition of a person to respond in a given way (i.e., to evaluate his/her ability to discriminate under experimental conditions Ec between the stimuli Sp and Sa) when certain experimental conditions are established in a second-order experiment. Thus, the terms D(P(Ec, Sp, Sa)) and I+(D(P(Ec, Sp, Sa))) designate dispositions, but they do not yet designate conscious experiences. The conscious visual experience of seeing a light stimulus is not a disposition but a subjective visual occurrence. To designate this conscious subjective experience, we use a logical calculus which includes an abstraction operator that creates abstract objects [54]. If L(a,b) means a loves b, then α (L(a, b) is the abstraction of L(a, b). α (L(a, b) denotes the love of a for b. I+(D(P(Ec, Sp, Sa))) represents “P can correctly evaluate his/her ability to discriminate under experimental conditions Ec between the stimuli Sp and Sa”, then α[I+(D(P(Ec, Sp, Sa)))] is the abstraction of I+(D(P(Ec, Sp, Sa))). As stated above, in this case, person P has privileged access (introspection) to his/her ability to discriminate the stimuli. α[I+(D(P(Ec, Sp, Sa)))] is the abstraction of this privileged access (introspection). If a person has such a priviledged access (introspection) to his/her ability to visually discriminate between stimuli, it can be said that this person has a conscious visual experience. We can express this with scientific precision by stating that the abstraction α[I+(D(P(Ec, Sp, Sa)))] designates the nature of the privileged access (introspection) or what we call in a non-scientific language the “conscious visual experience”. Everyone can experience how it is to have an experience that is designated by the the abstraction α[I+(D(P(Ec, Sp, Sa)))] when s/he is discriminating stimuli. 

In summary: D(P(Ec, Sp, Sa)) is a theoretical concept that is introduced with reduction sentences; I+(D(P(Ec, Sp, Sa))) is a theoretical concept that speaks about the theoretical concept D(P(Ec, Sp, Sa)), and is also introduced with reduction sentences; α[I+(D(P(Ec, Sp, Sa)))] is the abstraction of the theoretical concept I+(D(P(Ec, Sp, Sa))) which speaks about the theoretical concept D(P(Ec, Sp, Sa)). Although theoretical concepts cannot be defined with observational terms, they nevertheless play an indispensable role in scientific theories. However, not all psychological terms that are abstractions correspond to conscious experiences. In the case of “blindsight” without any conscious experience, a person cannot evaluate his/her ability to register the presence or absence of visual stimuli and has no privileged access to this ability. This can be expressed as I−(D(P(Ec, Sp, Sa))). In this case, the presence or absence of visual stimuli does not correspond to conscious visual experiences. The abstraction of the predicate “P cannot evaluate his/her visual ability to discriminate between the presence and absence of visual stimuli” then does not designate a conscious visual experience. From our own conscious experiences, we know which abstractions correspond to conscious experiences. We assume that other people have conscious experiences when they demonstrate the same responses under the same conditions, and when they demonstrate privileged access.

So far, only the two cases have been considered: A person can always detect the presence and absence of a stimulus or is unable to do so. In reality, there is often a smooth transition between these two extremes. When the visual ability of a person is impaired, the presence or absence of visual stimuli may only be detected with a given probability. This probability is investigated in the first-order experiment. This probability is represented by the index p1 which is inserted into the expression D(P(Ec, Sp, Sa)), resulting in the expression D^p1^(P(Ec, Sp, Sa)). p1 can take values between *p* = 0% and *p* = 100%.

To simplify the argument, only two results of experiment two were initially distinguished: (1) a person has no privileged access, i.e., α[I– (D^p1^(P(Ec, Sp, Sa)))] designates no conscious experience, and (2) a person has privileged access, i.e., α[I+(D^p1^(P(Ec, Sp, Sa)))] designates a conscious experience. If p1 indicates that the person cannot detect the presence or absence of visual stimuli in the first-order experiment, the stimulus cannot generate a visual experience. However, a second-order experiment can also determine the probability with which a person correctly evaluates his or her own visual ability. As described above, this estimation can be assessed in different ways: The least accurate and most questionable method is to ask the patient after each experiment whether s/he has seen anything. It is somewhat more accurate to ask the patient after each experimental trial whether s/he has seen anything. It is even more accurate to ask a person after each experimental trial to indicate on a rating scale how confident s/he is that s/he has perceived anything. The most accurate method is to have the patient compare the presentation of a stimulus in an area of the visual field where no processing of stimuli takes place with the presentation of a stimulus in an area of the visual field where visual stimuli are processed, as described above. The result is always the frequency with which the patient will provide a given response. This is expressed by the index p2 (replacing the indices + and −: α[I^p2^(D(P(Ec, Sp, Sa)))]). The result of the second-order experiment can take values between *p* = 0% and *p* = 100%. 

It must be demonstrated whether the result is due to chance or whether there is a significant relationship between the patient´s response in the second-order experiment indicating that a stimulus evoked a sensation when the stimulus was presented at a location where visual processing occurred. If the first-order experiment demonstrated that a patient can detect the presence or absence of visual stimuli (indicated by a significant p1 value), but if the second-order experiment did not yield a significant result (indicated by an insignificant p2 value), this demonstrates that the patient had no conscious experience when s/he detected the presence or absence of a stimulus. This corresponds to what has been termed “blindsight”. That a p1- or p2-value is “significant” means that it has been demonstrated that this value did not come about by chance but is due to the influence of the presence or absence of a visual stimulus. If p2 is very small, but if the result is significant, we can designate this in colloquial language as a very faint sensation. Increasing p2-values demonstrate an increasing distinctness of conscious sensation. One may term a significant result of a second-order experiment “type two blindsight” [32,33,34,35,36,37,38,39,40,41,42,43,44,45,46,47,48]. p2-values can express the range between unconscious processing of stimuli and conscous visual experience much more precisely than expressions such as “type one blindsight” or “type two blindsight” can do this. 

Taken together, the first-order experiment and the second-order experiment may have the following results: (1)p1 not significant: no visual experience;(2)p1 significant and p2 not significant: visual processing at a given level (represented by the value of p1) without conscious experience;(3)p1 significant and p2 significant: visual processing at a given level (represented by the value of p1) and conscious experience at a given level (represented by the value of p2).

The subjective experiences, when explicitly formulated, are abstract entities of mathematical logic, comparable to the wave equation for electrons in quantum mechanics. Whereas the presence of the wave structure of electrons can be demonstrated by the interference pattern on a screen, the presence of conscious experiences can be demonstrated by the behavior of a person under given conditions, the privileged access, and the simultaneous presence of neurobiological processes. A persons’ behavior under given conditions is intersubjectively observable, and the neurobiological processes can also be observed. For example, when a light stimulus hits a location on the retina (a receptive field), neurons receiving information from the retinal area corresponding to the receptive field, respond by increasing their electrical discharge rate, which can be visualized as an observable histogram. The question arises as to what the neurobiological foundations of conscious experience are and which impairments in cerebral functioning lead to blindsight.

## 6. The Neurobiological Basis of Conscious Vision and Blindsight 

### 6.1. Levels of Activation of Impaired Cortical Networks Resulting in Blindsight or Conscious Vision 

The phenomenon of “blindsight” may be due to residual functional neural networks in the damaged primary visual cortex. Numerous reports on the recovery of visual function in children [55,56,57] and adults [58,59,60] demonstrate that impaired neural networks can recover, and the patients may regain the ability to see in a formerly blind visual area. In addition, new connections may emerge and cerebral networks may rearrange. After cerebral hemispherectomy new fiber tracts can connect the ganglion cells of the blind retina to the remaining ipsilateral cerebral hemisphere [55]. Even in children devoid of the occipital lobe, the visual system can rearrange to such an extent that the children have normal luminance difference thresholds in the whole visual field [61]. The result of a previous experiment demonstrated that it may depend only on the degree of activation of a neural network whether a stimulus is not processed, whether there is only a feeling of the presence of a stimulus, whether a glimpse of a light is seen, or whether a light spot is clearly seen [36,37]. A 54-year-old man (RS) had a left homonymous hemianopia due to an embolic occlusion of the right middle and the right posterior cerebral artery and subsequent infarction of the right cerebral hemisphere. When a light spot (diameter: 69 min of arc; luminance: 101 cd/m^2^; background-luminance: 3.2 cd/m^2^) was moved within the affected (left) visual hemifield, four subareas of the visual field could be distinguished: an area where the patient was completely blind and where no visual processing occurred, an area in which the patient always reported the feeling of the presence of a stimulus without seeing anything, an area where the patient reported seeing a glimpse of light, and an area where the patient could see the light spot clearly. Three sessions of the visual field training were completed in 3 weeks. In each session, the light spot was presented at different locations in the left visual hemifield for 500 ms each. Scattering light was measured to exclude stray light artifacts. The visual field expanded after the conclusion of these three sessions. Now the patient could clearly see the light spot in the part of the visual field where he had previously seen only a glimpse of light. The patient could now see a glimpse of light in the visual area where previously he had only the sensation of the presence of a stimulus. In a part of the previously blind visual field, he now had the feeling of the presence of a stimulus without being able to see anything. After two months cessation of the visual field training, the visual field shrunk again and all the different areas returned to their positions before training. It is unlikely that within three weeks a reorganization of visual connections occurred, and disappeared again in the training-free interval. Therefore, it can be assumed that a neural network survived in the area of the primary visual cortex which represents the apparently blind visual field and that the patient´s experience depended on the extent to which the surviving tissue in the visual cortex was activated. This is in agreement with experiments that demonstrated that blindsight may be due to islands of activity in a damaged area V1 when stimuli are presented in the blind visual area. Using perimetry, Fendrich et al. [62] found islands of vision of which the patients were unaware in an apparently cerebrally blind area. Other authors have demonstrated areas of activation in a damaged primary visual cortex that represented a perimetrical blind visual area in which there were symptoms of blindsight [63,64] or in damaged areas of the primary visual cortex representing a perimetrically blind visual field [65,66]. 

### 6.2. Normal and Impaired Neural Networks in the Visual Cortex That Mediate Conscious Vision and Blindsight 

As stated above, a functional visual cortex is a necessary condition for normal vision in humans with a normally developed brain. Injury to the visual cortex can lead to blindness, and residual functions of a damaged cortex can be the neural basis for blindsight. Although clear differences exist between the cortical network of different mammalian species [67,68,69] most of the knowledge about the architecture and function of the visual cortex is based on examining the visual cortex of primates such as macaques and, in some cases, chimpanzees, because the greatest similarities were found between their cortices and those of humans [70,71,72,73]. The visual cortex of primates is divided in 6 layers, some of which have been divided into different sublayers [74]. Information from three different retinal ganglion cells (P-cells, M-cells and K-cells) predominantly reaches the 6- layered lateral geniculate nucleus (LGN) of the thalamus via the optic nerve [75,76,77]. At least 80% of ganglion cells in the fovea are P-cells. They convey information about the color and details of the shape of objects due to their high spatial frequency tuning and high visual resolution but transmit information slower than M-cells. M-cells mediate high temporal frequencies, and have lower spatial frequency tuning but higher conduction velocities than P-cells. They provide information about fast movements and high temporal frequencies but have large receptive fields and mediate low visual acuity [78,79,80]. The fibers of P-retinal ganglion cells terminate in the four dorsal layers of the LGN whereas the axons of M-cells terminate in the two ventral layers of the LGN [81]. The fibers of the geniculo-striate pathway end predominantly in area V1 of the visual cortex and to a much lesser extent in area V2, V3 [82,83,84,85], and V4 [86,87,88,89] (Figure 1). The magnocellular layers of the LGN project to area MT/V5 [90,91,92] and the inferior temporal gyrus including the lower bank of the superior temporal sulcus [93]. Neurons from these dorsal LGN layers project primarily to layer 4Cβ of area V1 and project to a much lesser extent to layers 6 and 4A and layer 1 of area V1 [94,95,96,97]. Layer 1 receives feedback input from layer 6 [98]. The P-cell projection to layer 4Cβ of area V1 constitutes approximately 18% of the synapses in layer 4Cβ. The highest density of synaptic contacts from the thalamus was found in this layer. A much lower rate of afferent thalamic input to spiny stellate cells was detected in layers 4Cα, which receives predominant input from M-cells, and layer 4Cβ. The density of synapses of thalamic afferents in layer 6 is approximately 16% of the density of thalamic afferent synapses in layer 4Cβ. M-neurons originating in the LGN project primarily to layer 4Cβ of area V1 [70,99]. K-ganglion cells of the retina project to thin koniocellular layers between the P- and M-cell layers of the LGN. There is a K-cell projection from the LGN to blob-like structures in layers 1 and 2/3 of area V1 and to area MT/V5 [100,101,102,103,104,105,106] (Figure 1). Layers that receive koniocellular input are also targeted by fibers from the superior colliculi [104,105,106]. The visual cortices of monkeys and humans contain a variety of neurons. Excitatory neurons can be divided into pyramidal cells and spiny stellate cells [107,108,109]. Spines are protrusions on the neurons’ dendrites. Interneurons were divided into different cell types such as Martinotti cells, horsetail shaped cells, neurogliaform cells, basket cells and chandelier cells. Inhibitory interneurons are located in all cortical layers. They are usually gamma-aminobutyric acid (GABA)-ergic, and may target the dendrites, the perisomatic region or the axons of pyramidal cells. Interneurons that contact the perisomatic region of pyramidal cells were called “basket cells”, and interneurons that target the initial segment of axons of pyramidal cells were termed “chandelier cells”. Neurons are further classified, but this classification is still a matter of debate [110]. Spiny stellate neurons in layer 4C project to layers 2–4B. The majority of 4Cβ neurons project to layers 2 and 3. There is a parvocellular and koniocellular input to layer 3Bβ [74]. Pyramidal cells and spiny stellate cells in layers 2–4B project to pyramidal cells in layer 5. Layer 4B is predominantly made up of pyramidal cells. The majority of layer 4B pyramidal cells project to area V2 whereas most neurons projecting to area MT/V5 are spiny stellate cells. A small number of pyramidal cells also project from area V1 to area MT/V5. Neurons that project to area V2 receive predominant input from M-neurons in layer 4Cα and from P-neurons in layer 4Cβ. Therefore, pyramidal neurons that project from layer V1 to layer V2 appear to integrate input from P-neurons via neurons of layer 4Cβ and input from M-neurons via neurons of layer 4Cα. Spiny stellate cells are a minority of layer 4B neurons. They receive their input exclusively from layer 4Cα which predominantly receives input from M-cells [111] (Figure 1). Neurons of layer 5 receive input from laminae 4B, 3B/4A, and 2/3A and a scarcer input from lamina 6. Neurons of lamina 5A project back to lamina 2/3A and laminae 3B/4A and 4C [112]. Area V1 is connected with Areas V2, V3, V3A, V4, MT/V5, the parieto-occipital cortex, and the posterior intraparietal cortex.

The contact between neurons is established by excitatory or inhibitory synapses. Excitatory synapses usually contact dendrites or dendritic spines of neurons. In contrast, inhibitory synapses mainly contact the soma and initial segment of neuronal axons and, to a lesser extent, spiny and non-spiny dendritic shafts. The boutons of presynaptic excitatory synapses contain the transmitter glutamate, which is the major transmitter in the primate brain, whereas the vesicles of inhibitory synapses are filled with gamma-aminobutyric acid (GABA) or glycine [113,114,115,116,117]. The postsynaptic membrane cytoplasm contains neurotransmitter receptors that modulate the effect of the transmitter released in the cleft between presynaptic and postsynaptic membranes [114,118]. When arterial blood flow is interrupted and neurons are no longer supplied with oxygen and nutrients, as was the case in many patients who exhibited blindsight (e.g., [18], patient 3 [22], patients SL, AG, EA [29], patient SL [50], patients AM, FB, LF [51], patients RC, PF, RA, JP [63], patient MC [64], the production of adenosine triphosphate (ATP) by oxidative phosphorylation, which is the predominant function of mitochondria, stops. As a consequence, the Na+/K+ ATPase pumps fail due to the lack of ATP, and Na+ accumulates inside and K+ outside the neurons. Glutamate is released and there is an increased inflow of calcium resulting in continuous neural discharge [119,120]. The cells either die or survive with severe injuries. In the former case, phagocytes remove dead cells, and over time the area is filled with a network of glia cells. In the latter case, neural metabolism is downregulated due to mitochondrial injury. The function of excitatory pyramidal cells and parvalbumin-positive GABAergic inhibitory interneurons that target the soma and dendrites of pyramidal cells is impaired, resulting in decreased inhibition of pyramidal neurons and impairment of the excitatory-inhibitory balance in the cortical network [121,122]. Mitochondria are transported along the axons to synaptic terminals and dendrites to provide energy at different locations, and are required for the synthesis of neurotransmitters, axonal transport, detoxification, regulation of calcium, ion gradient, and the organization of synaptic vesicles. After cerebral damage these mitochondrial functions may be impaired. Upon a decrease in arterial blood flow, blebs appear on the dendrites of the neurons, and part of the spines are lost [123,124]. The length of pyramidal dendrites decreases, and there is a loss of dendritic branches and shrinkage of the dendritic apical tree of (excitatory) pyramidal cells [125]. Thus, the function of neurons and the interconnections between neurons are impaired [126]. Traumatic brain damage, like in the patients described above (patient TU [36], patient FS (published several times [127,128,129], and patient GY, published many times, (see, e.g., [26,32,34,38,40,130,131,132,133,134,135,136,137], also results in cellular dysfunction and cell death due to cytotoxicity of blood, excitotoxicity, oxidative stress and inflammation, and the disruption of neural network, which results in a decrease in axon density [138,139,140]. Over time, surviving neurons can recover to some extent, new axons may grow and the function of spared neuronal networks may recover to some extent [141,142]. These results allow us to conclude that unconscious processing of visual stimuli (blindsight) and the feeling of the presence of a light stimulus without conscious visual experience are consequences of the loss of neurons, and their dendritic and axonal connections, as well as a downregulation of cell metabolism in the adult brain. If brain damage occurs in early childhood, a reorganization of neural connections and brain areas may occur, and neural networks of the visual system may emerge in brain areas that have other than visual functions in the normally developed healthy brain. This means that conscious visual experience requires a sufficiently activated, and sufficiently interconnected neural network that comprises a sufficient number of neurons with adequate metabolism. The availability of adenosine triphosphate (ATP) is a necessary condition for the function of the Na+/K+ pump, which is a prerequisite for an increased discharge rate of neurons when a visual stimulus appears in the receptive field of the neuron. The emergence of conscious experience requires the conduction of action potentials to other neurons in a neural network. Fast conduction of action potentials along axons requires a myelin sheath made up of oligodendrocytes and Schwann cells that wrap axons at regular intervals [143]. Demyelination causes loss of propagation of action potentials resulting in symptoms that include blurred vision or even blindness. The symptoms of demyelinating diseases demonstrate that the conduction of action potentials and interaction with other neurons is a necessary condition for the emergence of conscious experience ([144,145] for review). Synaptic connections between neurons play a crucial role in the emergence of conscious experiences. Conscious experiences are influenced by transmitters released in the synaptic gap and by experience-dependent dynamic modifications at the (mostly dendritic) postsynaptic membrane [146,147]. Overall, these structural and functional impairments in the primary visual cortex do not necessarily result in cerebral blindness. In some cases, the altered functioning of impaired neural networks may still mediate unconscious visual processing or the feeling that something is present without consciously seeing the target. If the impaired network of the primary visual cortex is highly activated, the visual experience of seeing a glimpse of light may emerge or the target may even be consciously seen. However, the geniculo-striate projection is not the only visual pathway. Therefore, the question arises as to whether a pathway other than the geniculate-striate projection can mediate blindsight.

### 6.3. Is Blindsight Mediated by the Secondary Visual Pathway? 

A PET study [130] demonstrated activation of area MT/V5 when the stimuli were moved at high velocity and when the patient reported conscious awareness of the stimuli. Thus, area V5 was activated in the assumed absence of activation of area V1. It was concluded that area V1 was not necessary for the conscious awareness of a rapidly moving stimulus. Fibers projecting from the LGN to area V5 bypassing area V1 may have mediated the visual perception of a moving stimulus in the absence of area V1.

**Figure 2 brainsci-12-01305-f002:**
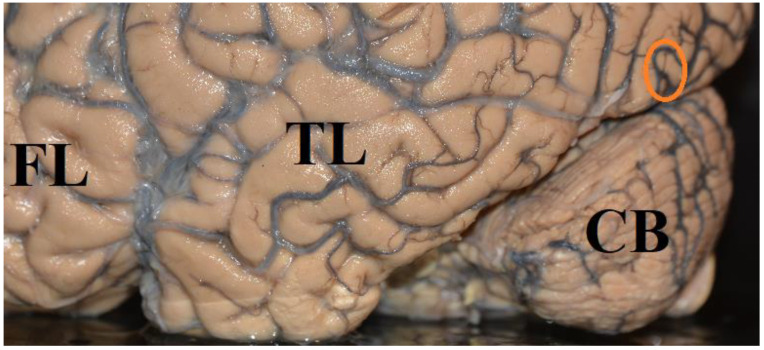
Location of area hMT+ (encircled) on the left hemisphere of the human brain. FL: frontal lobe, TL: temporal lobe, CB: cerebellum [148].

Experiments demonstrated activation of area hMT+ (Figure 2), when moving or stationary stimuli were presented in a visual area that was blinded due to a damaged primary visual cortex ([66,137,149,150,151,152,153,154] for reviews). In a study by Pedersini et al. [155], stationary or moving stimuli were presented in the cerebrally blind visual hemifield of 8 Patients with a homonymous visual field defect due to postgeniculate damage. The patients were asked to discriminate the orientation of a stationary or moving bar which was presented in the blind hemifield. Three out of 8 patients performed above chance in the moving condition and 4 out of 8 patients performed above chance in the static condition. Stimuli presented in the blind area activated visual areas V3, V4, hMT+ bilaterally, parietal and frontal areas and the insular and premotor cortex. The highest activation of area hMT+ was found for moving visual stimuli, of which the patient reported unawareness. Patients who performed above chance when discriminating moving stimuli without awareness demonstrated a higher activation of area MT+ than patients who performed at chance. Patients who performed better than chance in the static condition displayed higher activation of the contralesional area V1 and extrastriate visual areas. When pictures of human bodies and faces, butterflies, cars, or meaningless scrambles were presented to a patient who was blind due to the destruction of area V1 of both cerebral hemispheres, and when the patient was asked to guess (yes, no,) whether the stimulus belonged to a given category (e.g., a face), the patient reported no conscious visual experience for the stimuli. However, he reported seeing only changes of luminance when a new stimulus was presented. Nevertheless, the patient was able to guess the presence of human bodies above chance. Images of bodies activated the right extrastriate body area which is located in the lateral occipito-temporal cortex, in the vicinity of the motion-sensitive region hMT/V5+ [148,156,157], the right amygdala, orbitofrontal cortex, insula, superior temporal sulcus, and bilateral cerebellum. Pictures of faces activated the right gyrus cinguli, superior temporal sulcus, supramarginal gyrus, left superior parietal lobule, periaqueductal gray and the amygdala [158]. Barleben et al. [52] found activity in area hMT, the superior parietal lobule, supramarginal gyrus, and lateral and middle occipital gyri of 6 patients, whereas no activity was observed in the damaged striate cortex. In two patients who did not show activity in area hMT, lesions included subcortical pathways from the pulvinar to area hMT. None of the patients showed any symptoms of blindsight nor did they see rapidly moving stimuli (Riddoch effect). The authors conclude that there may be activity in area hMT, even though there is neither blindsight nor conscious visual experience. Some patients who participated in experiments on blindsight suffered from long-standing injuries dating back several decades and some patients (e.g., patient GY) participated in numerous experiments [26,28,32,126,127,128,129,130,131,132,133,134,135,136]. Over time, new connections could have developed and structures of the visual system could have been rearranged. Therefore, insights into the function of a damaged brain that has been stimulated in many experiments over many years should be applied with caution to normal brains. Repeated stimulation of cerebrally blind areas has been demonstrated to improve visual functions in a cerebrally blind hemifield [36,37,42,56,57,58,59,60] and can even lead to considerable restitution of visual functions due to a rearrangement of neural networks in children and adult patients [55,56,57,58,59]. 

Activation of area hMT+ can be mediated by a secondary visual pathway from the retina via the superior colliculi (SC) and the pulvinar. In monkeys, there is a direct projection from the retina to the SC [159,160,161,162,163,164,165]. About 10% of the retinal ganglion cells project to the SC [166]. Neurons of the SC were responsive to stationary and moving stimuli, size, color, and contrast [167,168,169,170,171,172]. The SC projects to the dorsal lateral geniculate nucleus and to the posterior and medial nuclei of the inferior pulvinar. The pulvinar is the largest nucleus of the primate thalamus and is divided into different subnuclei. The ventral pulvinar, which includes ventral parts of the lateral pulvinar and inferior pulvinar is connected with areas V1, V2, V4, and the inferotemporal cortex [173,174]. There is a rather sparse projection to the lateral and medial pulvinar and to the central lateral nucleus of the inferior pulvinar. The cortical area MT, which receives its main input from the medial nucleus of the inferior pulvinar does not receive input from the SC via the medial nucleus of the inferior pulvinar [175,176,177,178,179]. The SC projects to the caudal nucleus of the pulvinar and to the lateral and medial aspects of the rostral pulvinar. The caudal nucleus sends fibers back to the SC. The SC also projects to the dorsal and ventral parts of the lateral geniculate nucleus, to the pretectum, and to the inferior colliculi [180,181]. There is a retinotopic representation in the inferior and lateral pulvinar in rhesus monkeys. The inferior and lateral nuclei of the pulvinar respond to visual stimuli [182]. The results of a functional magnetic resonance imaging (fMRI) study in patients with bilateral lesions of the primary visual cortex demonstrate a direct connection between the LGN and area hMT+ which conveys visual information after damage to area V1 [151]. This study also demonstrates that the contralesional normal visual cortex is not needed for visual functions after damage to the primary visual cortex.

### 6.4. How Much Cortex Is Needed for Conscious Visual Perception?

The question is whether the SC, pretectum, and pulvinar are sufficient for mediating “unconscious” processing of visual stimuli. Georgy et al. [183] assumed that the SC (Figure 3) plays a pivotal role in processing gestalt-like or structured stimuli and in initiating motor responses. The authors investigated two hemispherectomized patients in whom the route from the retina to the SC on the hemispherectomized side was left intact. Stimulation of both visual hemifields yielded faster reaction times than single stimulation of the unaffected visual hemifield. The increasing speeds of the reaction times were especially pronounced if the stimuli were gestalt-like but not random shapes. However, this does not justify the assumption that the SC is sufficient to mediate the speeding of reaction times. In hemisperectomized patients, new unusual connections can develop over time, resulting in a projection from the visual hemifield contralateral to the hemispherectomy to the normal healthy cerebral hemisphere. Thus, the healthy hemisphere can represent both visual hemifields. Werth [55] has already demonstrated that light spots in the visual hemifield contralateral to hemispherectomy can be detected, localized, and reported as seen. In the patient who participated in this study, functional hemispherectomy was performed at the age of 135 months. The fibers targeting the frontal and occipital lobes were completely interrupted by undercutting the white matter underlying the frontal and occipital lobe. Light spots in the visual hemifield contralateral to the affected cerebral hemisphere were detected, locatized, and reported as seen up to 30 deg eccentricity. When light spots were presented in the affected left half of the visual field, functional MRI revealed activity in areas V1, V2, and V4 of the ipsilateral (left) hemisphere. These findings demonstrate that after hemispherectomy new fiber connections can be established contacting the occipital lobe of the hemisphere ipsilateral to the affected visual hemifield. Detailed investigation of visual functions in patients who underwend hemispherectomy in early life have demonstrated a rearrangemant of visual fiber connections to such an extent that stimuli were detected and localized up to 90 degrees eccentricity in the visual hemifield contraleteral to the missing cerebral hemisphere. Werth [55] reported the case of a 28-month-old child (patient FO) in whom the striate cortex and underlying white matter of the left cerebral hemisphere were replaced by a large cyst. Nevertheless, the child had a normally extended visual field with normal luminance difference thresholds in both visual hemifields. The child located light spots that were presented between 10 and 90 deg eccentricity, directed his gaze towards the stimuli and fixated them. Another patient (GI) [55] had undergone complete hemispherectomy at 4 months of age. This girl also had a normally extended visual field at the age of 59 months. The child detected light spots presented between 10 and 90 degrees eccentricity, directed eye and head movement towards the stimuli, and fixated them subsequently. However, the luminance difference threshold in the visual hemifield contralateral to the removed cerebral hemisphere was elevated. This demonstrates that a healthy hemisphere can represent both visual hemifields. It may be that the ability of “blindsight” in a visual hemifield contralateral to a removed cerebral hemisphere is also due to fiber connections targeting functional areas of the remaining cerebral hemisphere. However, children in whom the visual cortex and underlying white matter in both cerebral hemispheres are missing may still have a normally extended visual field with normal luminance difference thresholds. The child (KU) about whom Werth [55] reported, was a 19-month-old girl. The girl had developed a large prosencephalic cyst that was located in the occipital lobe of both cerebral hemispheres. The cyst included Brodman’s areas 17, 18, and 19 of both occipital lobes, both banks of the sulcus calcarinus, gyrus occipitotemporalis medialis, gyrus lingualis, and the cuneus and praecuneus. The child detected and located light spots between 10 and 90 deg eccentricity in both halves of the visual field, directed eye- and head movements to them and fixated them subsequently. 

### 6.5. Sensory Capacity of the Human Colliculi 

The capacity of the SC (Figure 3) to process sensory information can only be demonstrated in the complete absence of the telencephalon and preserved inferior and superior colliuli. Werth [61] reported the case of a 6-year-old hydranencephalic boy (patient AG), who regularly directed his head and eyes towards an auditory stimulus, although the child’s telencephalon was completely absent. Only the brainstem including the superior and inferior colliculi and the pretectum was preserved. In another child (patient HE), aged 28 months both cerebral hemispheres were replaced by a liquor filled cyst containing septum-like remnants of glial tissue. Only the brainstem including the pons, superior and inferior colliculi, the pretectum, and remnants of the ventral frontal lobe were preserved. The child was unable to locate light spots presented in her visual field, but regularly followed a face that was presented in the center of the visual field, with eye- and head movements [61]. If we assume that the presence of the telencephalon is a necessary condition for the emergence of a conscious visual or auditory experience, it could be said that these children also responded unconsciously to visual or auditory stimuli. This presumably unconscious processing of visual or auditory stimuli was mediated by the brain stem, including the inferior and superior colliculi and the pretectum. 

### 6.6. A Functional Visual System Is Not Sufficient for Conscious Visual Experience

If the projection from the retina to the cortex of the occipital lobe is unaffected by a cerebral lesion, and if the function of areas V1–V4 is unimpaired, conscious visual experience can nevertheless be absent. Patients who suffer from a neglect of one half of space after a cerebral lesion, demonstrate that an unimpaired primary and secondary visual system is not sufficient for conscious visual perception. They do not register objects in one half of space to the left or right of their body midline and do not recognize the left or right half of objects. They eat only from one half of a plate and do not notice that the plate has another half, and read only the text on the right half of a sheet and wonder about the incoherence of the text. If they are asked to draw an object, such as a flower, they draw only one half of the flower—usually the right half—and do not recognize that the other half is missing. They do not search for objects in the neglected half of space with eye and head movements, do not wash or dress one half of their body, and shave or apply makeup on only one half of their face [184]. A survey that also includes French and German literature ([185], for review), and reviews of the English literature [186,187,188,189] show that lesions causing a visual neglect of one half of space are predominately located in the the caudal parts of the supramarginal gyrus, angular gyrus, and the superior temporal sulcus of the right cerebral hemisphere. This demonstrates that conscious visual experience can only appear when many brain structures interact with the visual system.

## 7. Summary and Conclusions

In the present paper, it has been shown that the concepts “conscious visual experience” and “unconscious visual processing” can be introduced in a logically and methodologically correct way in the scientific language. Whether visual performance is classified as conscious or unconscious depends strongly on the experimental procedure used to draw the boundary between conscious and unconscious visual processing of stimuli. It turns out that patients’ claims of seeing nothing in a perimetrically blind visual hemifield and of only guessing the presence, orientation, shape, color or direction of motion of stimuli are not sufficient to determine whether a stimulus elicits a visual or other type of conscious experience. Unconscious processing of visual stimuli (blindsight), the feeling of the presence of a light stimulus, without conscious visual experience, and an elevated threshold for the emergence of conscious visual experience, are a consequence of the loss of neurons, and their dendritic and axonal connections, and downregulation of cell metabolism after damage to the visual cortex. Normal conscious visual experience requires a sufficiently activated number of interconnected neurons with sufficient metabolism. In the complete absence of the visual cortex, blindsight can also be mediated by a secondary visual pathway from the retina via the midbrain to cortical areas V4 and MT/V5.

## Figures and Tables

**Figure 1 brainsci-12-01305-f001:**
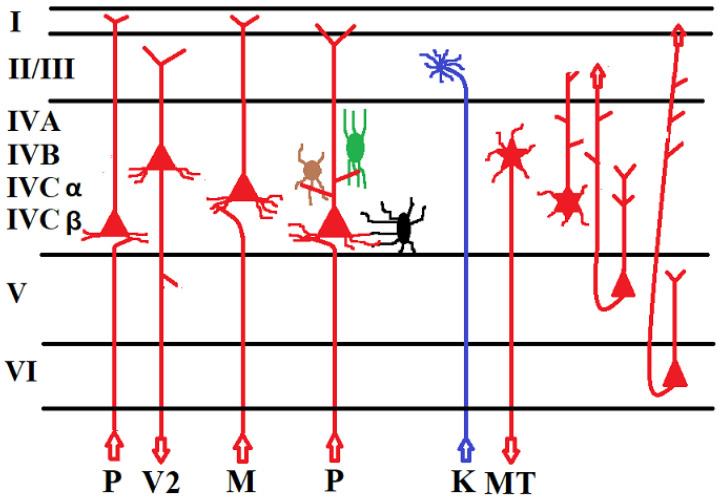
Schematic drawing of layers and cells in primate area V1. Red triangles: pyramidal cells with apical dendrites (dendritic spines are omitted). Red stars: stellate cells (dendritic spines are omitted). Black: basked cell contacting the perisomatic area of a pyramidal cell. Green: double bouquet cell contacting the apical dendrite of a pyramidal cell. Brown: chandelier cell contacting the dendrites of a pyramidal cell. Blue: konio cell (K) ending in a layer III blob. P: input from an LGN parvo cell contacting the dendrites of a pyramidal cell. M: input from an LGN magno-cell contacting the dendrites of a pyramidal cell. V2: axon of a layer IV pyramidal cell sending information to extrastriate visual area V2. MT: axon of a layer IV stellate cell sending information to area MT/V5.

**Figure 3 brainsci-12-01305-f003:**
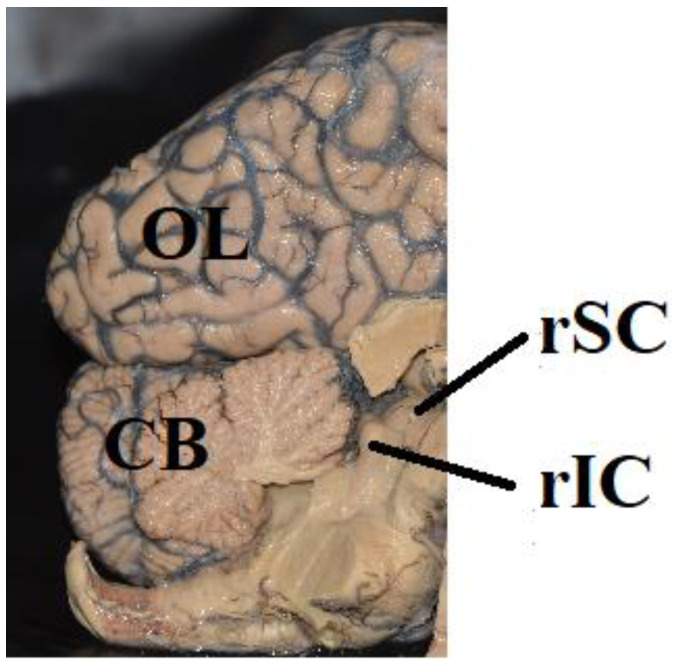
Right superior colliculus (rSC) and right inferior colliculus (rIC) of the human brain. OL: occipital lobe, CB: cerebellum.

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
