# Peer review of "A Scientific Approach to Conscious Experience, Introspection, and Unconscious Processing: Vision and Blindsight"

_brainsci, 2022, doi:10.3390/brainsci12101305_

Round 1
Reviewer 1 Report
In this review paper, the author demonstrates how the term conscious visual experience can be introduced logically and methodologically correctly and the distinction between conscious experience and unconscious processing in the cases of conscious vision and blindsight. Furthermore, the neurobiological basis of conscious vision and blindsight is discussed. I enjoyed reading this paper as it is clearly written and provides an excellent summary of the research on conscious visual experience and unconscious “vision”(blindsight). Boxes and figures are effective at illustrating complex concepts. This is a timely and relevant review, which will be influential in the field.
A minor suggestion is that it would guide the readers more effectively by adding references to the respective figures. These seem missing in the texts.
Author Response
The review of referee 1 is excellent and honors me. I have referred to all figures in the text, as
requested.

Reviewer 2 Report
This manuscript is a quite well written introduction to blindsight but it lacks depth and novelty. I cannot recommend publishing this manuscript in its current state. There are two ways to improve this: focus on making this the best systematic (framework based or at least somehow structured) review on blindsight so far or trying to bring a completely new angle to this topic like Danckert and Rossetti did in their review. As of now it tries to be little bit of both but it’s not working in my opinion.
Major concerns
In my view this manuscript lacks purpose. Who is it written for and what is it trying to achieve? It’s not a systematic review but it is not a proper standalone theoretical paper either. Some relevant references are missing. Consciousness researchers won’t find anything new from it but an it is not comprehensive enough to be an analysis on unconscious visual processing for broader audience. The most novel part in it is the denotation of unconscious visual processing, but I find it somewhat unnecessary complicated abstraction of this concept. It doesn't seem to fit to this manuscript and in my opinion it interrupts the flow. Also the proposed denotation doesn’t address the problem of discrimination between weak visual conscious experience and introspective access without visual experience. Both are conscious experiences but of different nature.
I don’t find the history of psychology/science part to be relevant. Introspection has been used in consciousness research for ages. Why is behaviorism brought up here? I feel like this discussion is over already and bringing it up makes it seem like there is an ongoing debate around introspection in consciousness research. Or at least non-report paradigms should be addressed here. If this section is absolutely necessary to keep I would mention Nagel as well as an example of criticism towards behaviorism:
Nagel, Thomas (1974). "What Is It Like to Be a Bat?". The Philosophical Review. 83 (4): 435–450. doi:10.2307/2183914.
Minor
The scale Mazzi and al. Were using is Perceptual awareness scale. You should mention original source: Ramsøy TZ, Overgaard M. Introspection and subliminal perception. Phenomenol Cognit Sci 2004;3:1–23.
Relevant research
Isa, T., & Yoshida, M. (2021, August 10). Neural Mechanism of Blindsight in a Macaque Model. Neuroscience. Elsevier Ltd. https://doi.org/10.1016/j.neuroscience.2021.06.022
Mazzi, C., Savazzi, S., & Silvanto, J. (2019). On the “blindness” of blindsight: What is the evidence for phenomenal awareness in the absence of primary visual cortex (V1)? Neuropsychologia, 128, 103–108. https://doi.org/10.1016/j.neuropsychologia.2017.10.029
H Railo, M Hurme (2021) Is the primary visual cortex necessary for blindsight-like behavior? Review of transcranial magnetic stimulation studies in neurologically healthy individuals Neuroscience & Biobehavioral Reviews 127, 353-364
Danckert, J., & Rossetti, Y. (2005). Blindsight in action: What can the different sub-types of blindsight tell us about the control of visually guided actions? Neuroscience and Biobehavioral Reviews, 29(7), 1035–1046.
Author Response
Referee 2: The referee´s major concern: I don’t find the history of psychology/science part to be
relevant. Introspection has been used in consciousness research for ages. Why is behaviorism brought
up here? I feel like this discussion is over already and bringing it up makes it seem like there is an
ongoing debate around introspection in consciousness research. Or at least non-report paradigms
should be addressed here. If this section is absolutely necessary to keep I would mention Nagel as well
as an example of criticism towards behaviorism:
Author: The referee´s assumption that Nagel presents arguments against behaviorism that refute it is a
complete misunderstanding of behaviorism and Nagel's paper. (I am not only a habilitated Dr. of
Medical Psychology but also a habilitated Dr. of Philosophy of Science, and as such I know what I am
talking about). Nagel repeats traditional positions that have been rejected by some philosophical
behaviorists, does not refute the arguments of behaviorists, and does not even quote them.
Behaviorists like the psychologist BF Skinner even agree with Nagel by assuming that subjective
experiences exist. For clarity, I inserted a short outline of Skinner´s behaviorism in the Introduction.
The behaviorist argument is not that subjecteive states do not exist, but that they do not play a role in
scientific language and do not explain behavior (Skinner 1953, 1974). Already Watson 1913, whom I
quoted in my paper, knew this. The problem is that in psychology and neuroscience up to now no
scientific term of „consciousness“ has been developed. Instead, the terms "conscious" and
"unconscious" are adopted in their unscientific everyday meaning. The authors assertion „I feel like
this discussion is over already and bringing it up makes it seem like there is an ongoing debate around
introspection in consciousness research“ demonstrates that the referee seems not to be aware of the
effort that has been made to make psychology a science. It appears that s/he does not understand the
necessity to scientifically specify the terms "conscious" and "unconscious", and considers the problem
superfluous. Without having a solution, the problem is simply ignored by many psychologists. In my
contribution I have demonstrated for the first time how these terms can be introduced logically and
methodologically correctly and how it can be distinguished experimentally correctly between
"conscious" and "unconscious" visual processing. In my view this is only possible using a logical
calculus with abstraction. This is even in agreement with Nagel´s view who writes „ … it should be
possible to devise a method of expressing in objective terms much more than we can at present, and
with much greater precision“ (Nagel 1974, p. 449). That is what I did in the present paper. I have
stated „From our own conscious experiences we know which abstractions correspond to which
conscious experiences. We assume that other people have conscious experiences when they
demonstrate the same responses under the same conditions, and when they demonstrate privileged
acces“ (Present paper, Section 5, first paragraph, last two sentences). This is in agreement with
Nagel´s view who writes „… we know what it is like to be us … we do not possess the vocabulary to
describe it adequately…“ (Nagel 1974, p. 440). Since we don't have the vocabulary, I developed it in
the language of mathematics (in a calculus with an abstraction operator). This is scientific and
objective. Each person can interpret what the abstraction designates for himself.
If I would quote Nagel´s paper, I would have to quote many other authors who have published on the
mind-body problem. Nagel's paper does not present an outstanding finding but repeats traditional
positions without offering a scientific solution. Therefore a citation of this paper would not be
appropriate.
Referee 2: In my view this manuscript lacks purpose. Who is it written for and what is it trying to
achieve?
Author: This is a reasonable request. Objectives and purpose were described in the revised version at
the end of the Introduction.
Referee 2: It’s not a systematic review but it is not a proper standalone theoretical paper either.
The reviewer does not say what his/her criteria are for what s/he calls a „systematic review“ and a
„proper stand alone paper“ and what is wrong with a paper that is not what the referee regards as a
„systematic review“ or a „proper stand alone paper“. In a review paper, important methods and results
should be described, concepts should be made precise and these issues should be discussed on the
grounds of logical inference. This is what my contribution does, irrespective of how the reviewer
designates it.
Referee 2: Consciousness researchers won’t find anything new from it but an it is not comprehensive
enough to be an analysis on unconscious visual processing for broader audience.
Author: The referee´s assertion plainly contradicts the facts: Some results discussed in the present
paper have only been published in German so far, others, published in Englisch, have not yet been
described and discussed in the blindsight literature. The following findings are new:
1) The concepts „conscious“ and „unconscious“ have been introduced for the first time in a logical and
methodological adequate manner.
2) It has been demonstrated that it is methodologically inappropriate to distinguish conscious and
unconscious visual processing of stimuli on the basis of patients' reports, and that the result depends
on the methods used. It is demonstrated how the distinction between "unconscious" and "conscious"
can be assessed experimentally and be replaced by a quantitative measure.
3) It is demonstrated that the extent of the visual field area in which a stimulus is reported as seen,
perceived only as a glimmer, felt as present, or reported as unregistered can vary within weeks, is due
to repeated visual stimulation, and appears to depend on the level of activation of the damaged visual
cortex.
4) It is demonstrated for the first time that after loss of one cerebral hemisphere, the remaining
hemisphere can mediate conscious visual functions in both visual hemifields,
5) It is demonstrated for the first time that after loss of the striate and prestriate cortex of one cerebral
hemisphere, the visual hemifield contralateral to the damaged hemisphere can have a normal extension
and normal luminance difference thresholds up to an eccentricity of 50 deg,
6) It is demonstrated for the first time that after loss of both occipital lobes during early development,
the visual field can have a normal extension and normal luminance difference thresholds,
7) It is shown for the first time that children with early loss of the occipital lobe of one cerebral
hemisphere, or who have been hemispherectomized, can exert normal saccades to stimuli in the visual
hemifield contralateral to the missing occitital lobe, or contralateral to the hemispherectomy.
8) It is demonstrated for the first time that auditory but not visual stimuli can elicit eye and head
movements towards the stimuli in a child without the telencephalon. This is the only study which
demonstrates that the superior colliculi are unable to mediate blindsight in humans.
The referee ignores all this and claims that the paper contains nothing new.
Referee 2: Also the proposed denotation doesn’t address the problem of discrimination between weak
visual conscious experience and introspective access without visual experience. Both are conscious
experiences but of different nature.
Author: The referee ignores or misunderstands what is said in the paper: It is clearly stated and
discussed that there is no introspection and no conscious experience if the rate of correct responses in
the second order experiment is at chance level. It is also explicitly stated and discussed that a result of
61 % correct may be regarded as as a „very low level of awareness“ (present paper, section 4. end of
third paragraph). Here the referee repeats what I have written. I think the referee agrees that if a result
does not differ from chance there is no sign of awareness.
Referee 2: The referee asserts that imprtant literature would be lacking.
This assertion is not a reasonable critique and is unjustified. The paper reviews the literature on
blindsight including 187 references. To my knowledge, all major literature is referenced. There exists
such a lot of literature that only the most important one can be cited. There is no paper in which all
existing literature is referenced. The assertion that some papers are not referenced is not a reasonable
criticism. The non-referenced literature, which the referee considers important, is not important from
my perspective. Nevertheless I have cited these papers to comply with the reviewer's request.
(References 53, 73, 158, 159). The missing papers are reviews which as such do not contain any new
experimental results. They do not report anything new that is missing in my contribution but results
and methodological issues that are reported and discussed in my contribution are missing in these
papers. For instance, how can a paper by Danckert and Rossetti that appeared 2005 contain more new
information than my present paper that includes 77 references which appeared after the paper by
Danckert and Rossetti? I will show in more detail that this reviewer´s assertion is without any
scientific justification and is in disagreement with the literature.
The paper of Danckert and Rossetti was published 2005, refers to earlier papers of Zihl and myself
(1984) but cannot include new results that have been publishing since. I have demonstrated for the
first time that after hemispherectomy new fiber connections to the remaining hemisphere can be
established, which even lead to conscious vision in the visual field contralateral to the
hemispherectomy. I have emphasized in the present paper that the question of whether the superior
colliculi may mediate blindsight (which Danckert and Rossetti designate as "action blindsight") can
only be answered if the entire telencephalon is lacking. I demonstrated that eye or head movements to
visual stimuli are not executed (no "action blindsight") in patients without a telencephalon whereas
eye and head movements to auditory stimuli are still possible. This has never been demonstrated
before and is completely new. I have described these results in the present review. Danckert and
Rossetti were unable to include a reference to these results and still assume that after hemispherectomy
the colliculi mediate "action blindsight". Thus, my explanations are much newer than the explanatioins
of Dankert and Rossetti. The reviewer's criticism is without justification.
The role of area MT/V5 which Danckert and Rossetti discuss is also discussed in my contribution. The
authors write on pp. 1037-1038: „ Given that recent fMRI research in humans demonstrated increased
activation in some areas of the human MT complex for both contralateral and ipsilateral motion
stimuli (Dukelow et al. ,2001) it may even be the case that this residual ability is supported, at least in
part,by the undamaged hemisphere. Further research would be needed to explore this possibility, as
well as examining in more detail the contribution of subcortical structures.“ This is exactly what I am
doing in the present paper refering to more recent studies including my own. The distinction that the
authors´ make between „action blindsight“ and „attention blindsight“ is not clear and the authors
suggest „Research … within the same patients will be needed to determine the extent to which
attention- and action-blindsight co-exist or co-vary“ (pp. 1038-1039). The authors regard form or
wavelength discrimination as „agnosopsia“ referring to Zeki and Ffytche 1998 who characterize this as
„not to know what one sees“ (p. 1038). This is by no means a precise scientific characterisation: the
words „know“ and „see“ are used in an unscientific every day language understanding, are no precise
scientific concepts and are not made precise by Danckert and Rossetti. The authors assume that „ …
an intact PPC (and perhaps a greater degree of sparing of dorsal extrastriate cortex) is needed for
action blindsight to be apparent in patients with hemianopia.“ (p. 140). I have demonstrated fort he
first time that patients without a telencephalon can orient to auditory stimuli and that
hemispherectomized patients can orient with eye and head movements towards visual stumuli in the
affected hemifield. These patients can also reach for light spots that appear in the visual hemifield
contraleteral to the hemispherectomy. This shows that neither PPC nor dorsal extrastriate cortex are
needed for orienting towards stimuli contralateral to the hemispherectomy. I could continue to show
that the paper by Danckert and Rossetti is an interesting review-paper for the publication year 2005.
But many new insights have been obtained since and the distinction between „action blindsight“,
„attention blindsight“ and „agnosopsia“ appears somewhat arbitrary in the light of today's knowledge.
That the referee highlights this work as most innovative and exemplary is inconceivable.
Referee 2
Referee 2: The referee advocates the paper „H Railo, M Hurme (2021) Is the primary visual cortex
necessary for blindsight-like behavior? Review of transcranial magnetic stimulation studies in
neurologically healthy individuals Neuroscience & Biobehavioral Reviews 127, 353-364“ as
important research.
Author: I quoted this paper (reference 53) in my revised paper because it demonstrates already in the
Introduction the importance of a clear characterization of the terms „conscious“, „unconscious“,
„visual perception“, and „experience“. The paper is a fine example for a scientifically absolutely
insufficient characterization of the term „conscious“: Railo and Hurme (2021) write: „We use the term
“conscious” vision to refer to visual perception that is accompanied by experiences that can be
introspected by the subject. We use “unconscious” visual perception to refer to situations where the
stimuli that the subject denies consciously seeing can nevertheless influence their behavior in some
way“ (Railo and Hurme 2021, page 354, first paragraph). An unclear term (conscious) cannot be
introduced by characterizing it using other unclear terms such as „visual perception that is
accompanied by experiences that can be introspected by the subject.“. What is the scientific meaning
of „visual perception that is accompanied by experiences“what is the meaning of „introspection“?
This is made clear in my paper refering to the state of the art in logic and methodology.
The authors repeatedly refer to the vague concepts „unconscious capacity“ „conscious vision“,
„degraded conscious vision „unconscious perception“ visibility“ without any effort to clarify these
concepts: „However, because many studies on blindsight patients have not employed systematic, trialby-trial investigations of visual experiences during the visual task, it is unclear how many instances of
reported unconscious capacity are merely severely degraded conscious vision (see also, Mazzi et al.,
2019; Overgaard, 2011). Studies by Overgaard et al. (2008) and Mazzi et al. (2016) show that what
may be initially interpreted as unconscious perception may just be degraded conscious vision, when
visibility is measured with graded scales.“ (p. 355, section 2.4, second paragraph) „Type-2 blindsight
refers to cases where the patients have some introspective access to the information about stimuli in
the scotoma.“(section 2.4, p. 355, last paragraph).
What is stated about the neurobiological grounds of blindsight is discussed in my paper with respect to
substantial literature that has been ignored in the paper by Railo and Hurme. Railo and Hurme´s
reasoning why TMS studies are needed is also not convincing. If one attempts to examine blindsight
before reorganization of cortical networks has begun and in whom the brain damage is localized,
patients after surgical removal of tissue in area V1 are more suitable than TMS studies in healthy
subjects. High resolution MRI can determine the extent of the injury in more detail.
In TMS studies revewed by Railo and Hurme, perception was measured only with a four or five step
visibility scale in which the subjects indicated if the stimulus could be seen. „Here, stimulus visibility
was measured using a four-step scale.“ (Section 4.1, first paragraph). These authors regard a
performance „unconscious“ when the subjects report lowest rate on the scale. The authors refer to the
study of Hurme et al. (2017) „In Hurme et al.’s study, “unconscious” was operationalized as follows:
the participants reported being confident (highest rating on three alternative scale) in seeing one
stimulus even though two were presented.“ In the same vein the authors write later „We conclude that
TMS studies provide evidence that information about the location or presence of a stimulus can guide
participants’ behavior, even when the participants report that they do not perceive the stimulus due to
V1 TMS.“ (Section 4.1, p. 358, last paragraph). The referee agrees with this very questionable way to
identify „unconscious“ processing of stimuli and with the inadequate way to „operationalize“ the
concept „unconscious“ as s/he recommends this paper. In my contribution I have shown by more
sophisticated experiments (comparing instances of no visual processing with instances where visual
processing occurred) that a person´s report whether or not s/he sees a stimulus is not suitable to
distinguish between conscious and unconscious visual processing.
It is also questionable whether TMS studies can „mimic“ blindsight. The authors themenselves state
that there is discrepancy between patients with cortical lesions and healthy subjects who have
undergone TMS: „We have shown that there is currently little evidence for strictly unconscious brain
stimulation induced blindsight in neurologically healthy observers. This means that there is
discrepancy between visual behavior observed in some patients with cortical lesions, and healthy
individuals when lesions are “mimicked” with non-invasive brain stimulation. This difference could
be explained by various methodological factors (e.g., small size of stimuli used in brain stimulation
studies).“ (p. 361, summary, section 5.4). Why then does the referee regard this paper as important for
blindsight, if it uses vage concepts (conscious, unconscious, experience, etc.) without making them
precise und when the authors themselves are reluctant to to compare TMS with blindsight? In the same
vein the referee criticises me for making vague concepts precise and for showing that it is insufficient
to distinguish between conscious and unconscious on the ground of the patients reports.
Referee 2: The paper by Mazzi, C., Savazzi, S., & Silvanto, J. (2019). On the “blindness” of
blindsight: What is the evidence for phenomenal awareness in the absence of primary visual cortex
(V1)? Neuropsychologia, 128, 103–108. https://doi.org/10.1016/j.neuropsychologia.2017.10.029 is
important but has not been referenced.
Author: The evidence on whether conscious visual awareness is possible without the visual cortex is
also reviewed in my paper. I also demonstrate what Mazzi et al. did not demonstrate : 1) that after loss
of one hemisphere the remaining hemisphere can mediate conscious visual functions in both cerebral
hemifields, 2) that after loss of the striate and prestriate cortex of one cerebral hemisphere the visual
hemifield contralateral to the damaged hemisphere can have a normal extension and normal luminance
difference thresholds up to an eccentricity of 50 deg, 3) that after loss of both occipital lobes in early
development the visual field can have a normal extension and normal luminance difference thresholds,
4) that children with early loss of the occipital lobe of one cerebral hemisphere or who have been
hemispherectomized can exert normal saccades to stimuli in the visual hemifield contralateral to the
missing occipital lobe or contralateral to the hemispherectomy, and 5) that the results of my studies
contradict the assumption that blindsight can be mediated by the human superior colliculi alone.
Referee 2: The scale Mazzi and al. Were using is Perceptual awareness scale. You should mention
original source: Ramsøy TZ, Overgaard M. Introspection and subliminal perception. Phenomenol
Cognit Sci 2004;3:1–23.
Relevant research: Isa, T., & Yoshida, M. (2021, August 10). Neural Mechanism of Blindsight in a
Macaque Model. Neuroscience. Elsevier Ltd. https://doi.org/10.1016/j.neuroscience.2021.06.022
Author: I have cited these two papers to satisfy the referee´s requests (references 73 and 159).

Round 2
Reviewer 2 Report
This manuscript has potential but unfortunately it still falls short in my opinion. The author failed to address my conserds due to his dismissal or misunderstanding of my critique. The author has made minimal changes to the manuscript and the previously mentioned conserns hold.
Some clarifications:
If this paper is a review it should state the methodology used. How was the literature search conducted and what were the inclusion criteria? These need to be justified.
Dismissing the fenomenology of blindsight is a major issue and it seems that the author doesn't really understand the deeper message of Nagel's paper.
The utility of the formulation is questionable. Just because something is expressed in the language of mathematics doesn't make it more scientific or true. Now the deeper problem "Each person can interpret what the abstraction designates for himself" is assigned to the interpreter but the same problems still exisist. How this formulation handles different taxonomies of blindsight should be discussed.
Author Response
How this formulation handles different taxonomies of blindsight should be discussed.
Author: This is explained in datail in the following additional text (sectiom 5, second paragraph from
below):
To simplify the argument, only two cases were initially distinguished: 1) a person has no privileged
access, i. e. α[I-
(D(P(Ec
, Sp, Sa)))] designates no conscious experience, and 2) a person has privileged
access i. e. α[I+
(D(P(Ec
, Sp, Sa)))] designates a conscious experience. However, a second-order
experiment can also determine the probability with which a person correctly evaluates his or her own
visual ability. As described above, this estimation can be determined in different ways: The least
accurate and most questionable method is to ask the patient after each experiment whether s/he has
seen anything. It is somewhat more accurate to ask the patient after each experimental trial whether
s/he has seen anything. It is even more accurate to ask a person after each experimental trial to indicate
on a rating scale how confident s/he is that s/he has perceived anything. The most accurate method is
to have the patient compare the presentation of a stimulus in an area of the visual field where no
processing of stimuli takes place with the presentation of a stimulus in an area of the visual field where
visual stimuli are processed, as described above. The result is always the frequency with which the
patient will provide a given response. This is expressed by the index p (replacing the indices + and -:
α[Ip
(D(P(Ec
, Sp, Sa)))]). The result of experiment 2 can take values between p= 0% and p=100% . It
can be demonstrated whether the result is due to chance or whether there is a significant relationship
between the response indicating that a stimulus evoked a sensation when the stimulus was presented at
a location where visual processing occurred. If it is demonstrated that experiment 2 did not yield a
significant result, indicating that the patient had no conscious experience when a stimulus appeared,
this corresponds to the assumption that the person had no conscious experience even though visual
stimuli, were processed, This was termed „blindsight“. If p is very small, but if the result is significant
we can designate this in colloquial language as a very faint sensation. Increasing p-values demonstrate
an increasing distinctness of sensation. One may term a significant result of experiement 2 " type two
blindsight" [32-48]. p-values can express the range between unconscious processing of stimuli and
conscous visual experience much more more precisely than expressions such as „type one blindsight“
or „type two blindsight“ can do this.
If this paper is a review it should state the methodology used. How was the literature search
conducted and what were the inclusion criteria? These need to be justified.
Author: The following text has been added (last 6 lines of section 1.):
“The review is based on several thousand publications about the anatomy, physiology, and
neuropsychology of the visual system in humans which were available in the Max-Planck-Institute for
Psychiatry, the Bavarian State Library, the Library of the Medical Faculty of the University of
Munich, Pubmed, Science Direct, Psycnet or other internet-based databases, which the author
collected over about 40 years up to the year 2022. 190 studies that were considered the most relevant
to the questions posed in the present review were included.”
Dismissing the fenomenology of blindsight is a major issue and it seems that the author
doesn't really understand the deeper message of Nagel's paper.
Author: When Nagel presents his perspective, he can do so with empirical data and with
logically correct inferences. This is the way to convey a message. Besides this, there is no
„deeper message“ as referee 2 claims. Instead of discussing the behaviorist arguments and
possible counterarguments and my arguments concerning Nagel´s paper, referee 2 refers to a
mysterious „deeper message“. I explained why Nagel´s paper is not an argument against
behaviorism. Again: To disprove philosophical behaviorism one must demonstrate that
subjective experiences can be designated introducing concepts that designate these private
experiences without refering to conditions and responses under which these responses occur.
This means that the criteria that specify whether the designation is correct must be private.
This implies a Wittgensteinian private language which is regarded as impossible because such
an assertion would imply a logical contradiction. Nagel does not adress this core problem of
philosophical behaviorism and he presents no solution. The position of psychological
behaviorism is that private experiences may exist but they play no role in scientific
psychology. To disprove this it must be demonstrated that there are psychological concepts
that cannot be introduced on the grounds of behavior or physiological reactions and which
enable scientifically testable assertions that cannot be made with concepts that are introduced
by reactions under testing conditions (I added (section 1, paragraph 4) an outline of
Skinner´s behaviorism demonstrating that even radical behaviorism that Skinner advocated
does not contradict Nagel´s view). Nagel does not adress this problem in his paper. Referee 2
appears to be completely unaware of the behaviorist arguments and refuses to present a any
argument. Instead, referee 2 tries to tell us that I misunderstood Nagel, again without
presenting a single argument. If referee 2 alleges that I misunderstood Nagel, then he must
demonstrate where I have misunderstood Nagel, he must demonstrate where Nagel quotes
behaviorist positions in his paper and where he refutes behaviorist arguments. The referee is
unable to do this because Nagel doesn't discuss this in his paper. He does not deal with the
arguments of behaviorist philosophers, and does not deal with psychological behaviorism. He
does not even mention the advocates of these positions. Therefore, to assert that I did not
understand Nagel´s paper is nothing but polemic. The referees remarks demonstrate that s/he
is a complete layman (laywoman) in this field.
The author failed to address my conserds due to his dismissal or misunderstanding of my
critique. The author has made minimal changes to the manuscript and the previously
mentioned conserns hold.
Author: I could not dismiss or misunderstand the referee´s critique because referee 2 does not
present a single scientifically sound argument that one could dismiss or misunderstand. A
„critique“ is a logically sound argument that contradicts claims. Why doesn´t the referee show
me where he presents a logically correct argument that I dimissed.
The only reason to change a paper is that there are arguments that contradict the assertions
made in the paper, or one can add text if constructive suggestions are made to supplement the
text. I mentioned already in my reply that referee 2 did not present a single logically sound
argument that contradicts a claim that I made. In his/her reply (round 2) referee 2 also does
not demonstrate which argument he presented against a claim that I made. The only
reasonable suggestion referee 2 made is to request a clear description of the objectives and
scope of the paper. This has been added on section 1, end of second paragraph from below.
Author: The referee´s question „In my view this manuscript lacks purpose. Who is it written
for and what is it trying to achieve?“ has been answered. The purpose and the scope of the
paper has been described on page .
Author: I have clearly disproved the referee´s comment „Consciousness researchers won’t
find anything new from it but an it is not comprehensive enough to be an analysis on
unconscious visual processing for broader audience“. Therefore I have listed 8 items that are
completely new in my paper (see my response). However, ther referee ignores this.
The I have made clear that the referee´s assertion „Also the proposed denotation doesn’t
address the problem of discrimination between weak visual conscious experience and
introspective access without visual experience. Both are conscious experiences but of
different nature.“ is not a counterargument because the referee only repeats what I have stated
in the paper.
I have argued in detail why the referee´s claim that imprtant literature would be lacking is
baseless. I have argued that many results that I have discussed in my paper are missing in
these papers, that no new results are reported, and that no concepts are clarified. Why does
referee 2 ignore this?
The utility of the formulation is questionable. Just because something is expressed in the
language of mathematics doesn't make it more scientific or true.
Author: This is also no scientific argument but a hollow phrase without content. Referee 2
claims The utility of the formulation is questionable without demonstrating what is
questionable. S/he is unable to show that the logical calculus I use is inadequate or that the
logical rules which I use to introduce the concepts „unconscious“ and „different degrees of
consciousness“ are violated, or that the experimental procedure is inadequate. Referee 2
refuses to accept a logically exact specification of the terms „conscious“ and „unconscious“
without presenting arguments, and simoultaneosly s/he recommends papers that have made
completely absurd scientific attempts to introduce these terms. I have demonstrated this in
detail in my last response to referee 2. I have even discussed this paper that the referee regards
important in my paper showing how insufficient the approach that is made in this paper is.
But referee2 simply ignores all this.
Assertions become indeed more scientific when sophisticated facts that cannot be expressed
in normal language are expressed mathematically. "More scientific" means, that facts are
described more precisely and quantitatively and calculation rules and the rules of logical
inference can be applied. Does referee 2 really think that the Schrödinger equation is not a
scientific advance in quantum mechanics or that tensor algebra is not an important
contribution to the General Theory of Relativity?
No one ever claimed that an assertion is true when translated into the language of
mathematics. In this way assertions can be unambiguously formulated and quantitative
predictions can be made, which would be impossible without a translation into mathematics.
Only after psychological findings are translated in the laguage of statistics it can be proven
that a result is significant and is different from chance. The truth of these assertions can then
be investigated empirically. Almost the entire technical world is based on mathematically
formulated classical physics, quantum mechanics and theory of relativity. Does referee 2
really want to assert that mathematics is dispensable?
Also the exploration of the boundary between unconscious processing of stimuli and possible
different levels of conscious experience requires quantitative concepts (see: section 5, second
paragraph from below) that should be described as clear as possible and that should be
formulated in a semantically unambiguous language which does not lead to
contradictions(i.e., logic). I have still stated in the paper the unscientific ambiguous concepts
of "conscious" and "unconscious" that have been used in papers on blindsight.
Now the deeper problem "Each person can interpret what the abstraction designates for
himself" is assigned to the interpreter but the same problems still exisist.
Author: This is also by no means a scientific argument. Referee 2 does not say what the
deeper problem is, which problems still exists and why they exist. If referee 2 alludes to the
problem how to designate conscious experiences in a semantically clear and in a logically
adequate way his/her assertion is mistaken. The referee ignores again what has been discussed
in detail in my paper. The problem how to designate conscious experiences in a semantically
clear and in a logically adequate way has been described in detail in the paper, and a solution
has been proposed. Thus the problem how to designate conscious experiences in a
semantically clear and in a logically adequate way does not exist any longer. If referee 2
asserts that problems still exist s/he must say which problems still exist. But s/he does not do
this.
